# Risk of Human Pathogen Internalization in Leafy Vegetables During Lab-Scale Hydroponic Cultivation

**Gina M. Riggio [1], Sarah L. Jones [2] and Kristen E. Gibson [2],***

[1]   Cellular and Molecular Biology Program, Department of Food Science, University of Arkansas, Fayetteville, AR 72701, USA; gmriggio@email.uark.edu

[2]   Department of Food Science, University of Arkansas, Fayetteville, AR 72704, USA; slj017@email.uark.edu

*   Correspondence: keg005@uark.edu; Tel.: +1-479-575-6844

**Abstract:** Controlled environment agriculture (CEA) is a growing industry for the production of leafy vegetables and fresh produce in general. Moreover, CEA is a potentially desirable alternative production system, as well as a risk management solution for the food safety challenges within the fresh produce industry. Here, we will focus on hydroponic leafy vegetable production (including lettuce, spinach, microgreens, and herbs), which can be categorized into six types: (1) nutrient film technique (NFT), (2) deep water raft culture (DWC), (3) flood and drain, (4) continuous drip systems, (5) the wick method, and (6) aeroponics. The first five are the most commonly used in the production of leafy vegetables. Each of these systems may confer different risks and advantages in the production of leafy vegetables. This review aims to (i) address the differences in current hydroponic system designs with respect to human pathogen internalization risk, and (ii) identify the preventive control points for reducing risks related to pathogen contamination in leafy greens and related fresh produce products.

**Keywords:** hydroponic; leafy greens; internalization; pathogens; norovirus; *Escherichia coli*; *Salmonella*; *Listeria* spp.; preventive controls

## 1. Introduction

In 2018, the United States (U.S.) fresh produce industry was implicated in three separate multi-state outbreaks linked to contaminated field-grown romaine lettuce from Arizona and California, which produce 94.7% of the leafy greens in the U.S. [1]. The three leafy green outbreaks were cited in 294 illnesses and six deaths across the U.S. [2–4]. From 1973 to 2012, leafy greens have comprised more than half of the fresh produce-associated outbreaks reported in the U.S. [5]. While risk management strategies and regulatory requirements (e.g., the Food Safety Modernization Act Produce Safety Rule) were developed in response to produce-associated outbreaks, these are primarily applicable to conventional, field-grown crops as opposed to controlled environment agriculture (CEA). Meanwhile, CEA is a growing industry and a potentially desirable alternative production system, as well as a risk management solution for the fresh produce industry. According to a 2017 survey of over 150 farms worldwide, a total of 450,000 square feet of production space was added during a one-year period [6]. Moreover, 16% of responding farms had opened during that same one-year period [6].

For hydroponic systems to be a viable risk management strategy for addressing food safety issues in the leafy vegetable industry, established CEA producers that use hydroponics must strive to balance productivity with produce safety. Currently, there are minimal science-based reports on the benefits of CEA overall with respect to product safety. Moreover, although conventional production systems have made great strides through the adoption of Good Agricultural Practices (GAPs; e.g., Leafy Greens Marketing Agreement), traditional field growers may look to CEA and hydroponics as an opportunity

to enhance the safety of their product along with the longevity of their operations. This review aims to (i) address the differences in current hydroponic system designs with respect to human pathogen internalization risk, and (ii) identify preventive control points for reducing the risks related to pathogen contamination in leafy greens and related fresh produce products.

*Review Methodology*

To inform this review paper, the authors searched the following databases: Web of Science, PubMed, and Google Scholar. The key word search terms were a combination of the following: foodborne pathogens, food safety, pathogen internalization, endophytic, hydroponic, soilless, soil-free horticulture, greenhouse, indoor farm, growth chamber, leafy greens, lettuce, leafy vegetables, microgreens, and herbs. Additional searches were done for specific human pathogens, including Shiga toxin-producing *Escherichia coli*, *Salmonella enterica*, *Listeria monocytogenes*, human norovirus, and its surrogates Tulane virus and murine norovirus. The authors further narrowed the search for studies in hydroponic systems by searching for the names of specific types of systems such as deep water culture, wick systems, nutrient film technique, continuous drips, as well as the phrases 'flood and drain' and 'ebb and flow'. Numerous studies have been conducted on pathogen internalization in fresh produce as reviewed by Erickson [7], and these studies include all of the production systems and produce types, as well as experimental designs investigating internalization outside of the 'normal' germination process (e.g., directly through stomata as opposed to roots). For the present review, studies were excluded if they did not specifically study internalization via roots, if they did not include a technique resembling soilless horticulture, or if they were investigating internalization in produce that are typically eaten raw and were not leafy vegetables (e.g., tomato, cantaloupe, or berries). Based on these criteria, 17 papers were identified for primary discussion in Section 5.

## 2. Controlled Environment Agriculture (CEA) and Food Safety

Controlled environment agriculture encompasses a variety of non-traditional farming methods that take place inside climate-controlled buildings. Examples of CEA locations may include greenhouses or high tunnels, which have transparent or translucent walls that let in natural sunlight. CEA may also include indoor buildings or warehouse spaces with opaque walls that rely on artificial lighting for photosynthesis. Greenhouses and fully indoor spaces require varying degrees of climate modulation, such as heating, cooling, humidity control, $CO_2$ injection, and supplemental lighting. Indoor farmers often use soil-free horticultural techniques including hydroponics, aquaponics, aeroponics, or growing on mats (e.g., Biostrate) and soil alternatives (e.g., coco coir). This review will focus on hydroponic leafy vegetable production (including lettuce, spinach, microgreens, and herbs), which can be categorized into six types: (1) nutrient film technique (NFT), (2) deep water raft culture (DWC), (3) flood and drain, (4) continuous drip systems, (5) the wick method, and (6) aeroponics [8,9]; however, aeroponics will not be discussed in this review. Overall, each of these systems may confer different risks and advantages in the production of leafy vegetables.

A 2016 survey of 198 indoor farms by Agrilyst [10], an indoor farm management and analytics platform company, reported that 143/198 (72%) of farms grow leafy greens, herbs, or microgreens, and 98/198 (49%) of respondents use hydroponic greenhouses as their operating system. Furthermore, 86% of the small CEA farms (<1500 square feet) stated that they planned to expand their farm size "over the next five years," as stated in the survey question [10]. Previous research on food safety practices on small to medium-sized field-based farms demonstrates that these groups typically struggle to maintain consistent food safety practices [11,12]. If these trends are similar to indoor hydroponic farmers, it will be imperative to deter inadequate food safety practices in beginner CEA growers before they expand. In general, a preventive control point of particular concern in fresh produce production is agricultural water quality. While numerous studies, as reviewed by De Keuckelaere et al. (2015), have investigated the impact of agricultural water quality on the food safety aspects of field-grown crops [13], very little attention has been given to their CEA counterparts. In hydroponic leafy vegetable farming, pathogen

internalization via contaminated nutrient solution could be a significant issue as well as an obvious control point; thus, more detailed research in this area is needed for developing relevant guidelines.

Furthermore, because hydroponic systems are often housed in built environments, pathogens may more feasibly recirculate in air handling systems and in the recirculating water supply. Microbiome studies of the built environment infrastructure suggest that humans are the main driver of microbial diversity in these settings, and a wide variety of microbes occupy niches in the buildings [14]. Additionally, human handling can contribute significantly to the contamination of fresh produce [15]. Human pathogens commonly associated with contaminated fresh produce include *Listeria monocytogenes*, *Salmonella enterica* serovars, Shiga toxin-producing *E. coli* (STEC), and human noroviruses, which are the most common cause of gastroenteritis associated with fresh produce [16–18]. Each of these pathogens has characteristics that enable their survival in the built environment for weeks to months or even years [19–21]. The presence of persistent microorganisms within the environment could lead to the superficial deposition or even internalization of pathogens in leafy vegetables.

## 3. Pathogen Internalization in Leafy Vegetables

Internalization refers to the transfer of microorganisms from the environment to the inner tissue of the plant. One of the earliest studies demonstrating pathogen internalization in fresh produce was Hara-Kudo et al. [22]. The study was in response to a July 1996 outbreak in Sakai City, Japan involving hydroponically grown radish sprouts contaminated with *Escherichia coli* O157:H7 that sickened ~6000 people [23]. Hara-Kudo et al. [22] demonstrated that contamination of either the seed or hydroponic water with *E. coli* O157:H7 can result in marked colonization of the edible parts of the sprout. In addition, the frequency of internalization increased with increasing concentrations of *E. coli* O157:H7 in the hydroponic water. Meanwhile, Itoh et al. [24] used immunofluorescence microscopy and scanning electron microscopy to confirm pathogen contamination on the surface, in leaf stomata, and on inner plant tissue such as xylem. The internalization of *E. coli* O157:H7 in lettuce cut edges has also been observed, even following chlorine treatment [25]. In one of the first field trials, Solomon et al. [26] demonstrated that soil (i) fertilized with *E. coli* O157:H7-contaminated manure or (ii) irrigated with contaminated water both led to the internalization of *E. coli* O157:H7 in the lettuce tissue, as confirmed by fluorescence microscopy. Since internalized pathogens cannot be effectively removed by post-harvest disinfection [27], a large body of research has been conducted in order to address the mechanisms, causes, and prevention of pathogen internalization in fresh produce, specifically leafy vegetables.

It is well established, as shown in lab-based experiments, that foodborne pathogens can become internalized and disseminated in plant crops via the plant root systems, through wounds in the cuticle, or through stomata, as shown in lab-based experiments [28–30]. Multiple reviews have thoroughly addressed the pathogen internalization of leafy vegetables. Hirneisen et al. [30] concluded that internalization is specific to the plant and pathogen, and that the use of soil or hydroponic media highly impacts the absorption of microorganisms in produce. The authors go on to conclude that healthy, non-injured roots appear to hinder the internalization of microorganisms, and that if an uptake of pathogens does occur, the microbial load does not directly correlate with the concentration in leaves and stems. Hirneisen et al. [30] determined that, in general, pathogen internalization within the edible portion of leafy greens was observed less frequently in contaminated soil-based systems compared to contaminated hydroponic systems. In studies where internalization was greater in soil, it was attributed to root damage during growth [31] or features of soil, such as resident microorganisms, that may suppress internalization through competition [31,32]. Other reviews support the notion that hydroponic systems pose a greater internalization risk [7,32–34] with water as a common source of contamination [35]. Therefore, it is critical to identify contamination risk factors within the various hydroponic plant culture systems and define potential preventive control measures for hydroponic leafy vegetable growers.

## 4. Hydroponic System Designs

Hydroponic crop production combines irrigation and fertilization into one system by submerging plant roots in buffered fertilizer salt solutions. Hydroponic plant culture systems and the terminology used to describe them vary widely. However, there are some common design themes such as the use or non-use of a solid horticulture substrate, active pumping or passive water flow, open-cycle or closed-cycle water use, the degree to which the roots are submerged in water, the method of root aeration, and whether the flow rate is zero, continuous, or intermittent (Table 1). These characteristics are potentially relevant to pathogen internalization via roots because they determine the nature of the physical contact between the plant root system and the nutrient solution.

The five systems most commonly described in the literature for growing leafy vegetables include the NFT, DWC, flood and drain, continuous drip [36], and the wick method [37]. Aeroponics, where roots are sprayed with a nutrient solution rather than submerged, can also be used for leafy vegetables. However, the aeroponics technique was developed primarily for growing root crops for the herbal supplement industry [38], and thus will not be discussed in this review. Hydroponic systems may also be classified by the container type used, such as window boxes, troughs, rails, buckets, bags, slabs, or beds [36,39]. For the purpose of this review, they have been grouped by how the roots interact with the nutrient solution (Figure 1).

The preparation of seedlings for hydroponic systems includes germination and transplantation. Germination is usually performed by adding one seed to a piece of a moistened solid medium called a "plug", which is often made of rockwool, or a netted cup filled with peat and perlite. Plugs must be stabilized with a nutrient solution of pH = 4.5–5.6, sub-irrigated, and then germinated for 2–3 weeks at 17–20 °C under a humidity dome. For NFT systems, it is of particular importance that the roots penetrate the bottom of the plug before transplanting, so that they can extend into the nutrient solution [39–41].

**Table 1.** Hydroponic leafy vegetable systems compared to conventional farming systems.

| Deep Water Raft Culture | Nutrient Film Technique | Continuous Drip | Wick Method | Flood and Drain | Conventional, Field-based |
|---|---|---|---|---|---|
| **Submergence of plant roots in nutrient solution** | | | | | |
| Roots are fully submerged in NS throughout the growing process. | Root tips touch a 1–10-mm film of NS running along the bottom of plastic gutters. | Roots grow through a solid matrix in a grow bed that is filled with NS. | Roots are fully submerged in NS throughout the growing process. | Roots grow through a solid matrix in a grow bed that is mostly filled with NS when flooded, and exposed to air when not flooded. | Roots are fully covered by the soil matrix and exposed to water through irrigation. |
| **Water Flow** | | | | | |
| No water flow | NS is actively pumped continuously or intermittently at a low flow rate. | NS is actively pumped continuously at a low flow rate. | No water flow in plant reservoir. NS is passively replenished through capillary action from the tank up through fibrous wicks. | Grow bed is periodically flooded with NS at a higher flow rate than NFT or drip, by active pumping, and then drained. The pump is typically timer-controlled. | Roots grow in soil and are watered by drip irrigation and surface watering. |
| **Water recirculation** | | | | | |
| OC | CC | CC | OC | CC | OC |
| **Solid phase** | | | | | |
| No | No | Yes | Yes | Yes | Soil, compost, manure |
| **Method of root aeration** | | | | | |
| Injection | All but the root tips are exposed to the air inside the gutters. | Agitation from pump | Injection | Exposed to air during drained periods, from agitation by the pump during flood periods. | By ensuring adequate soil drainage |

Solid phase = Y: Gravel, perlite, vermiculite, pumice, expanded clay, plastic mats, plastic beads, rice hulls; NFT, nutrient film technique; NS, nutrient solution; OC, open-cycle; CC, closed-cycle; Soil, silt loams, sandy soils, or clay with good drainage.

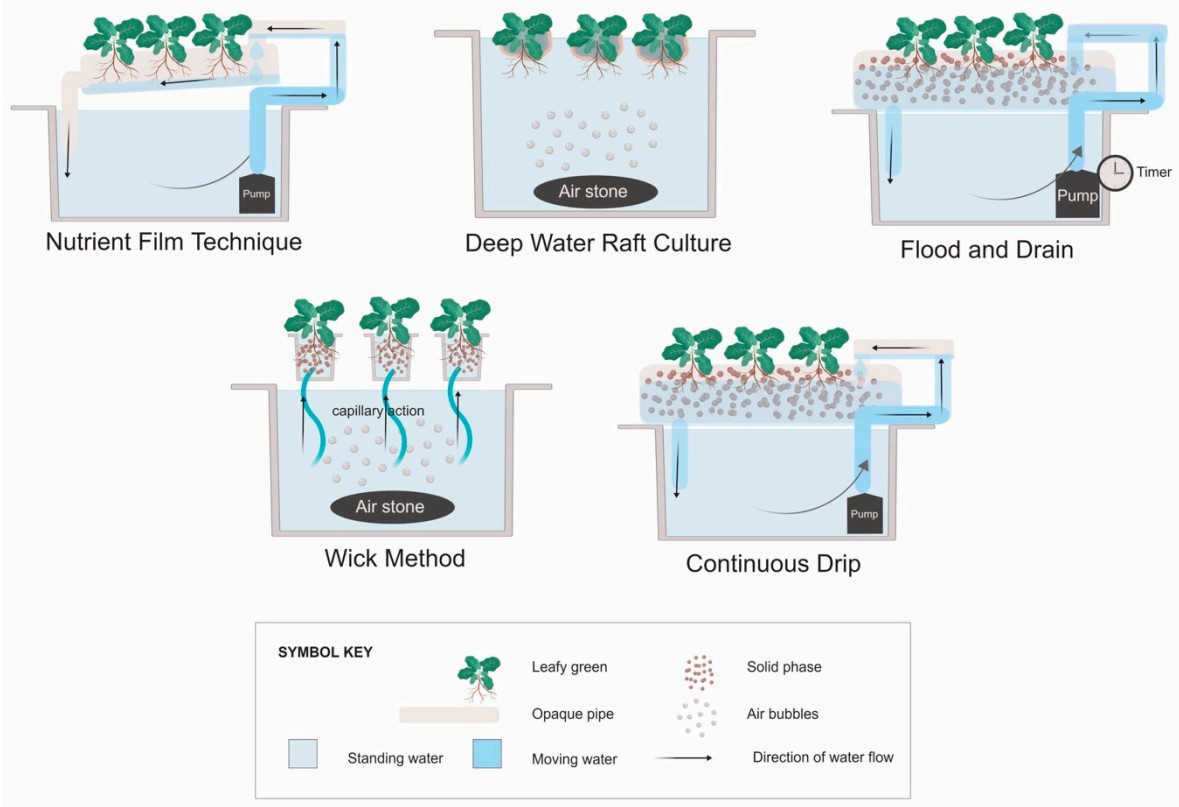

**Figure 1.** Types of hydroponic plant culture systems. "Deep water raft culture" may also be referred to as "float hydroponics" [36], while "flood and drain" can be referred to as "ebb and flow" [39]. The "continuous drip" system is typically called a "drip system" [36], but "continuous" is used here to differentiate it from flood and drain systems that have similar construction, but the pump runs intermittently.

By contrast, the planting process for commercial field-based lettuce production is most often seeded directly into the soil using pelleted seeds and a mechanical seeder; however, an increasing minority of lettuce crops is transplanted. Generally, seedlings that are used for transplant are 4–6 weeks old, sowed in 200-well seed trays, and germinated at a target temperature of 20 °C. Most irrigation is performed by surface drip [42–45].

## 5. Pathogen Internalization in Hydroponic Systems

Few studies involve hydroponic systems that are representative of commercial operations. Laboratory-scale plant cultivation resembling the hydroponic concept dominates the literature, using Hoagland's solution in trays, tubes, or flasks. This method is similar in concept to deep water culture, as no pumps, recirculation, or aeration are typically used, and the roots are mostly or fully submerged in the solution [31,46–49]. In some lab-based systems, plants were cultivated using an agar-solidified hydroponic nutrient solution rather than a fluid solution. Two studies have utilized a NFT or NFT-like system [50,51], while one study utilized a continuous drip system, but inoculated the solid phase as opposed to the nutrient solution [52]. Research addressing the internalization of pathogens in leafy vegetables across a variety of hydroponic systems has been summarized in Table 2.

**Table 2.** Investigations of pathogen internalization in leafy greens grown hydroponically by system type.

| System Type | Solid Phase | Pathogen | Plant | Inoculation | Surface Sterilized | Compared with Soil | Internalization Outcome | Ref. |
|---|---|---|---|---|---|---|---|---|
| HA-GB | N/A | *E. coli* O157:H7, *Salmonella* Typhimurium, and *L. monocytogenes* | Carrot, cress, lettuce, radish, spinach and tomato | Seeds soaked in 2 log CFU/mL, and then air-dried on sterile filter paper for 2 h at ~22 °C | Yes | No | Levels of all pathogens increased from 2 log to ~5–6 log CFU during 10-day germination. Counts and SEM showed a plant-specific effect (cress and radish most susceptible), a pathogen-specific effect (*L. monocytogenes* most abundant), and an age-specific effect (internalization was greater in young plants) | [28] |
| DWC-L-T | No | *E. coli* TG1 expressing GFP | Corn seedlings (*Zea mays*) | 7 log CFU/mL added directly to the 4-L tray of nutrient solution | No | No | Internalized *E. coli* TG1 detected in shoots. Entire root system removed (430 CFU/g), root tips severed (500 CFU/g), undamaged plants (18 CFU/g). | [29] |
| DWC-L-F | No | GFP-expressing *E. coli* O157:H7 and *S.* Typhimurium (MAE 110 and 119) | Lettuce (*Lactuca sativa* cv. Tamburo) | 29 mL of hydroponic nutrient solution with a final concentration of 7 log CFU/mL | Yes | Yes | Hydroponic: *S.* Typhimurium MAE 119 internalized at 5 log CFU/g. | [31] |
| DWC-L-T | No | GFP-expressing *E. coli* O157:H7 from a spinach outbreak and a beef outbreak as well as a non-pathogenic clinical *E. coli* isolate | Spinach | 3 and 7 log CFU/mL or g added directly to the nutrient solution or soil. Group 1: Inoculated hydroponic for 21 d; Group 2: Hydroponic for 21 d, transplanted into sterile soil; Group 3: hydroponic for 21 d, transplanted into inoculated soil | Yes | Yes | At both 4 log and 7 log CFU/mL in hydroponic water, between 2–4 log CFU/shoot internalized pathogen detected at cultivation day 14. Soil recovery was negligible for both high and low inocula and required enrichment to detect. 23/108 soil-grown plants showed *E. coli* in root tissues, but no internalization in shoots. | [32] |
| DWC-L-F | Sand | *S.* Typhimurium (LT1 and S1) and *L. monocytogenes* sv4b, *L. ivanovii*, *L. innocua* | Barley (*Hordeum vulgare*) | 8 log CFU/mL suspension per bacterial species added directly to the surface of the sand 1 to 2 days after planting | Yes | No | *Salmonella* internalized in roots, stems, and leaves, while *Listeria* spp. only colonized the root hairs. | [46] |
| DWC-L-C | No | Six strains of *E. coli* O157:H7, five strains of *S.* Typhimurium and *S.* Enteritidis, six strains of *L. monocytogenes* | Spinach (*Brassica rapa* var. perviridis) | 3 or 6 log CFU/mL added directly to the hydroponic water solution | No | No | Across all microorganisms, the 3 log CFU/mL had an average recovery of <1.7 log CFU/leaf in 7/72 samples. The 6 log CFU/mL inoculum resulted in better recovery (50/76 samples) in a range of 1.7 to 4.4 log CFU/leaf. | [47] |

**Table 2.** *Cont.*

| System Type | Solid Phase | Pathogen | Plant | Inoculation | Surface Sterilized | Compared with Soil | Internalization Outcome | Ref. |
|---|---|---|---|---|---|---|---|---|
| DWC-L-T | No | *E. coli* O157:H7 | Spinach cultivars Space and Waitiki | 5 or 7 log CFU/mL added directly to the Hoagland medium. Hoagland medium was re-inoculated as needed to maintain initial bacterial levels. | Yes | Yes | *E. coli* O157:H7 internalized in 15/54 samples at 7 days after inoculation with 7 log CFU/mL. Neither *curli* or spinach cultivar had an impact on the internalization rate. | [48] |
| DWC-L-J | Vermiculite | *Coxsackievirus* B2 | Lettuce (L. sativa) | 7.62–9.62 log genomic copies/L in water solution | Unknown | No | Virus detected in leaves on the first day at all inoculation levels; however, decreased to below LOD over the next 3 days. | [49] |
| NFT | Rockwool plugs | *E. coli* P36 (fluorescence labeled) | Spinach (*Spinacia !oleracea* L. cv. Sharan) | 2 to 3 log CFU/mL E. coli added to the nutrient solution in the holding tank. 2 log CFU/g was added to soil. | Yes | Yes | For hydroponic: total surface (7.17 ± 1.39 log CFU/g), internal (4.03 ± 0.95 log CFU/g). For soil: total surface (6.30± 0.64 log CFU/g), internal (2.91± 0.81 log CFU/g) | [50] |
| NFT | No | MNV | Kale microgreens (*Brassica napus*) and mustard microgreens (*Brassica juncea*) | Nutrient solution containing ~3.5 log PFU/mL on day 8 of growth | Unknown | No | MNV was internalized into roots and edible tissues of both microgreens within 2 h of nutrient solution inoculation in all samples at 1.98 to 3.47 log PFU/sample. After 12 days, MNV remained internalized and detectable in 27/36 samples at 1.42 to 1.61 log PFU/sample. | [51] |
| DS | Peat pellets/clay pebbles | MNV (type 1), *S.* Thompson (FMFP 899) | Basil (*Ocimum basilicum*) | MNV (8.46 log-PFU/mL) or S. Thompson (8.60 log-CFU/mL) via soaking the germinating discs for 1 h | No | No | MNV was internalized into edible parts of basil via the roots with 400 to 580 PFU/g detected at day 1 p.i., and the LOD was reached by day 6. Samples were positive for *S.* Thompson on days 3 and 6 post-enrichment. | [52] |
| DWC | No | *Citrobacter freundii* PSS60, *Enterobacter* spp. PSS11, *E. coli* PSS2, *Klebsiella oxytoca* PSS82, *Serratia grimesii* PSS72, *Pseudomonas putida* PSS21, *Stenotrophomonas maltophilia* PSS52, *L. monocytogenes* ATCC 19114 | Radish (*R. sativus* L.) microgreens | Final concentration of 7 log CFU/mL for each bacterium added directly to the nutrient solution | Yes | No | *C. freundii* PSS60, *Enterobacter* spp. PSS11, *K. oxytoca* PSS82 were suspected to have internalized in hypocotyls. These three strains were detected with and without the surface sterilization of plant samples. | [53] |
| HA-TT | N/A | *Klebsiella pneumoniae* 342, *Salmonella* Cubana, Infantis, 8137, and Typhimurium; *E. coli* K-12, E. coli O157:H7 | Alfalfa (*M. sativa*) and Barrelclover (*M. truncatula*) | 1 to 7 log CFU/mL added directly to the growth medium at the seedling root area after 1 day of germination. | Yes | No | *K. pneumoniae* 342 colonized root tissue at low inoculation levels. *S.* Cubana H7976 colonized at high inoculation levels. No difference between *Salmonella* serovars | [54] |

**Table 2.** *Cont.*

| System Type | Solid Phase | Pathogen | Plant | Inoculation | Surface Sterilized | Compared with Soil | Internalization Outcome | Ref. |
|---|---|---|---|---|---|---|---|---|
| HA-TT | N/A | *S.* Dublin, Typhimurium, Enteritidis, Newport, Montevideo | Lettuce (*Lactuca sativa* cv. Tamburo, Nelly, Cancan) | 10 µL of a 7 log CFU/mL suspension per serovar added directly to the 0.5% Hoagland's water agar containing two-week old seedlings | Yes | Yes | Hydroponic: *S.* Dublin, Typhimurium, Enteritidis, Newport, and Montevideo internalized in *L. sativa* Tamburo at 4.6 CFU/g, 4.27 CFU/g, 3.93 CFU/g, ~3 CFU/g, and ~4 log CFU/g, respectively | [55] |
| DWC | No | hNoV GII.4 isolate 5 M, MNV, and TV | Romaine lettuce (*Lactuca sativa*) | TV and MNV (6 log PFU/mL), and hNoV (6.46 log RNA copies/mL) added directly to the nutrient solution | Yes | No | TV, MNV, and hNoV detected in leaves within 1 day. At day 14, recovery levels were TV: 5.8 log PFU/g, MNV: 5.5 log PFU/g, and hNoV: 4 log RNA copies/g were recovered | [56] |
| DWC | Vermiculite | *E. coli* O157:H7 | Red sails lettuce (*Lactuca sativa*) | Started with 7 log CFU/mL and maintained in water at 5 log CFU/mL | Yes | No | *E. coli* O157:H7 internalized in contaminated lettuce of cut and uncut roots. Mean uncut: 2.4 ± 0.7; Mean 2 cuts: 4.0 ± 1.9; Mean 3 cuts: 3.3 ± 1.3. No significant difference was found between two and three cuts. | [57] |
| DWC-(AP) | Vermiculite | Total coliforms | Red sails lettuce (*Lactuca sativa*) | No inoculation. Detected 2 to 4 log CFU/mL natural concentration of coliform bacteria in a pilot system downstream of a cattle pasture | Yes | No | UV light at 96.6% transmittance and a flow rate of 48.3 L/min reduced total coliforms by 3 log CFU/mL in water. Internalized coliform was not recovered from either samples or control lettuce. | [58] |

AP, aquaponics; C, cups; CFU, colony-forming units; DS, drip system; DWC, deep water culture; DWC-L, DWC-like; GB, grow beds; GFP, green fluorescent protein; HA, hydroponic agar; hNoV, human norovirus; J, jars; LOD, limit of detection; MNV, murine norovirus; NFT, nutrient film technique; PFU, plaque forming units; p.i., post-inoculation; SEM, scanning electron microscopy; T, trays; TT, test tubes; TV, Tulane virus.

Briefly, Table 2 is designed to highlight the key aspects impacting the microbial internalization results of the lab-scale hydroponic studies, including the type of microorganisms, plant type and cultivar, inoculation procedure, and the application of surface sterilization prior to microbial analysis. With respect to surface sterilization, 12 out of the 17 studies cited in Table 2 specifically described the application of a decontamination procedure prior to microbial recovery and detection. Most of the investigators validated the decontamination procedures and showed the complete inactivation of external microorganisms while maintaining the viability of internalized microorganisms.

### 5.1. Deep Water Culture

DWC systems are the most prominent hydroponic CEA systems used, thus making them of heightened interest to researchers [59]. As outlined in Table 1, DWC systems traditionally do not have a solid phase component, and yet many studies use a DWC-like system that does include various solid phase components (Table 2). Therefore, for the purposes of this review, DWC-like systems without a solid phase will be compared here, while those with a solid phase are discussed in Section 5.3.

In a traditional DWC system, Settanni et al. [53] used a variety of microorganisms (Table 2) to inoculate the hydroponic solution for radish microgreen cultivation. To determine if internalization occurred, researchers sampled the mature hypocotyls of the plants, and found that less than half of the microorganisms were found to be internalized and in "living form" in the plant tissue. *Citrobacter freundii*, *Enterobacter* spp., and *Klebsiella oxytoca* were found to have internalized within the hypocotyls. These three strains were detected with and without the surface sterilization of plant samples, indicating microbial persistence both externally as well as via internalization.

Macarisin et al. [48] used a DWC-like system with no solid phase to grow two spinach cultivars. The researchers inoculated *E. coli* O157:H7 into the hydroponic medium and soil to study the impact of (i) curli expression by *E. coli* O157:H7, (ii) growth medium, and (iii) spinach cultivar on the internalization of the bacteria in plants. Curli are one of the major proteinaceous components of the extracellular complex expressed by many *Enterobacteriaceae* [60]. When curli fibers are expressed, they are often involved in biofilm formation, cell aggregation, and the mediation of host cell adhesion and invasion [60]. Neither the curli expression by *E. coli* O157:H7 nor the spinach cultivar impacted internalization. The authors found that under experimental contamination conditions, spinach grown in soil resulted in more internalization incidences when compared to those grown hydroponically. These data highlight that injuring the root system in hydroponically grown spinach increased the incidence of *E. coli* O157:H7 internalization and dissemination throughout the plant. The authors concluded that these results suggest that *E. coli* O157:H7 internalization is dependent on root damage and not the growth medium, which could be linked to (1) root damage in soil or (2) increased plant defenses in hydroponics where plants were exposed to repeated contamination.

Similar to Macarasin et al. [48], Koseki et al. [47] utilized hydroponically cultivated spinach to determine potential pathogen internalization. Briefly, the authors inoculated hydroponic medium at two concentrations (3 and 6 log colony-forming units [CFU]/mL) with various strains of *E. coli* O157:H7, *S.* Typhimurium and Enteritidis as well as *L. monocytogenes*. The authors observed that the 3 log CFU/mL inoculum resulted in limited detection (seven out of 72 samples) of internalized bacteria with an average concentration of <1.7 log CFU/leaf (i.e., limit of detection of the assay) across all bacteria. The 6 log CFU/mL inoculum level resulted in greater detection (50 out of 76 samples) ranging from >1.7 to 4.4 log CFU/leaf.

Meanwhile, Franz et al. [31] inoculated their hydroponic nutrient solution with 7 log CFU/mL of *E. coli* O157:H7 and *S.* Typhimurium (MAE 110 and MAE 119). The two morphotypes of *S.* Typhimurium, MAE 110 and 119, represent a multicellular phenotype with the production of aggregative fimbriae and a wild-type phenotype lacking the fimbriae, respectively. The internalization of *S.* Typhimurium MAE 119 in the leaves and roots of lettuce Tamburo occurred at approximately 5 log CFU/g, while *E. coli* O157:H7 did not result in any positive samples, thus indicating that internalization likely did not occur. Additionally, *S.* Typhimurium MAE 110 was only detected at an average of 2.75 log CFU/g in roots.

The lack of internalization by the MAE 110 type within the hydroponic system was an interesting finding, as it was previously suggested that the aggregative fimbriae are critical in the attachment and colonization of plant tissue [61]. Finally, similar to Macarasin et al. [48], Franz et al. [31] hypothesized that *E. coli* O157:H7 must be more dependent on root damage for the colonization of plant tissues, as significant differences in internalization were observed between hydroponic and soil-grown lettuce, with the latter more likely to cause root damage.

Interestingly, the study by Klerks et al. [55] also documented serovar-specific differences in the endophytic colonization of lettuce with *Salmonella enterica*, as well as significant interactions between *Salmonella* serovar and lettuce cultivar with respect to the degree of colonization (CFU per g of leaf). More specifically, the root exudates of lettuce cultivar Tamburo were reported to attract *Salmonella*, while other cultivars' root exudates did not. These authors utilized a hydroponic agar system, which is discussed further in Section 5.3.

Sharma et al. [32] reported one of the few studies that directly compared the hydroponic and soil cultivation of spinach. The researchers determined that there was no detectable internalization of *E. coli* in spinach cultivated in the soil medium. In comparison, 3.7 log CFU/shoot and 4.35 log CFU/shoot of *E. coli* were detected in shoot tissue from all three replicate plants grown in inoculated hydroponic solution on days 14 and 21, respectively. The authors suggested that the semisolid nature of the hydroponic solution may have allowed motile *E. coli* cells to travel through the medium more readily when compared to soil. In addition, populations of *E. coli* increased in the hydroponic solution over time, while the soil population levels declined to less than 1 log CFU/g by day 21. This difference is likely due to the lack of environmental stressors on *E. coli* cells in the hydroponic solution, which improves the internalization capacity in spinach tissues.

DiCaprio et al. [56] investigated the internalization and dissemination of human norovirus GII.4 and its surrogate viruses—murine norovirus (MNV) and Tulane virus (TV)—in romaine lettuce cultivated in a DWC system. Seeds were germinated in soil under greenhouse conditions for 20 days prior to placement in the DWC system with feed water. The feed water (800 mL) was inoculated with 6 log RNA copies/mL of a human norovirus (hNoV) GII.4 or 6 to 6.3 log plaque-forming unite (PFU)/mL of MNV and TV to study the uptake of viruses by lettuce roots. Samples of roots, shoots, and leaves were taken over a 14-day growth period. By day 1 post-inoculation, 5 to 6 log RNA copies/g of hNoV were detected in all of the lettuce tissues, and these levels remained stable over the 14-day growth period. For MNV and TV, the authors reported lower levels of infectious virus particles (1 to 3 log PFU/g) in the leaves and shoots at days 1 and 2 post-inoculation. MNV reached a peak titer (5 log PFU/g) at day 3, whereas TV reached a peak titer (6 log PFU/g) at day 7 post-inoculation. The authors suggested that it is possible that different viruses may have varying degrees of stability against inherent plant defense systems, thus explaining the variation amongst the viruses within this study, as well as other studies on this subject.

*5.2. Nutrient Film Technique*

While NFT is more commonly used by small operations, the NFT production share is growing [62]. If contaminated hydroponic nutrient water is capable of introducing pathogens via plant roots—and the roots of NFT-grown plants make contact with the nutrient water only at root tips—it is worth investigating if this reduced root surface contact (i.e., compared to DWC) has an impact on pathogen internalization risk. If differences are identified, system choice could be added to food safety guidelines for indoor-grown leafy greens, and would have no such analogous recommendation in soil-based production guidance. Unfortunately, at the time of this review, only two studies have been published that address pathogen internalization using the NFT for hydroponic leafy green production (Table 2).

Warriner et al. [50] compared non-pathogenic *E. coli* P36 internalization in hydroponic spinach and soil-grown spinach. For spinach grown in contaminated potting soil, *E. coli* P36 was detected consistently from day 12 to day 35 post-inoculation on leaf surfaces at concentrations of 2 to 6 log CFU/g. However, *E. coli* P36 was not detected internally in roots or leaves until day 32 at

~2 log CFU/g. Meanwhile, 16 days post-inoculation, ~2 log CFU/g of *E. coli* P36 were detected in and on roots, but not leaves. Both soil and NFT nutrient water had a starting concentration of 2 log CFU/mL of *E. coli* P36. These data suggest that *E. coli* P36 internalizes poorly overall in soil-grown spinach, and preferentially internalizes in the roots of hydroponic spinach. This is supportive of the hypothesis that motile bacterial species may be a greater risk in hydroponic systems than in soil. However, these results differ from the findings reported by Franz et al. [31] and Macarisin et al. [48] with respect to the role of motility in the *E. coli* O157:H7 colonization of plant tissues cultivated in hydroponic systems.

A separate study demonstrated that MNV spread throughout a NFT system that had been used in the cultivation of kale and mustard microgreens [51]. After inoculating the nutrient solution with 3.5 log PFU/mL of the virus on day 8 of cultivation, viral RNA was detected at $10^4$ to $10^5$ copies per 10-g microgreen sample, and internalized virus was detected at 1.5 to 2.5 log PFU per 10-g microgreen sample. Similar levels were observed in roots and edible parts. Levels of virus in the nutrient water lingered at ~2 log PFU/mL for up to 12 days. Moreover, the authors demonstrated cross-contamination to the second batch of microgreens at 2 log PFU/sample of internalized virus.

These two studies suggest that both bacteria and viruses are capable of internalizing in leafy greens within NFT systems, and to a greater degree than soil for bacteria [50]. However, non-standard measurements and different starting inoculum concentrations between studies make true comparisons difficult. For example, at both 4 log and 7 log CFU/mL contamination of hydroponic water in a DWC system, between 2–4 log CFU per spinach shoot of internalized *E. coli* O157:H7 was detected after day 14 of cultivation. By contrast, Warriner et al. [50] detected ~2 log CFU/g of internalized *E. coli* after 16 days of cultivation, but it is difficult to compare "grams" and "shoots" without knowing the weight of the shoots, which was not reported. Additionally, it is unknown if certain *E. coli* strains internalize more effectively than others. Indeed, species-specific and strain-specific differences have been reported [28,31,46,55].

The paucity of data related to NFT systems and the pathogen contamination of leafy greens suggest that more research is needed. In particular, the standardization of NFT systems for research purposes needs to be pursued. For instance, Warriner et al. [50] suggested that the rockwool plugs used for seed germination and subsequent cultivation in their NFT system may have had a filtering effect, as evidenced by the *E. coli* levels dropping in the system over time while increasing in soil. If the rockwool plugs were submerged sufficiently to absorb contaminants, this may not have been a true NFT system, as only the root tips should touch the water. It may also indicate that hydroponic systems that use a solid phase (Figure 1) are at increased risk for internalization via root systems due to the accumulation of contaminants in the growth medium during recirculation. Since only the plant root tips are typically submerged in the contaminated nutrient solution in NFT, but internalization is similar, perhaps the root tips are principle routes of entry for human pathogens. Plant root cell division and elongation occurs at the greatest extent at root tips and also at root junctions [63], possibly leaving ample opportunity for pathogen entry. However, as data accumulate, it may be revealed that NFT systems do not differ from DWC production with respect to pathogen internalization risk.

*5.3. Other Hydroponic Systems*

While DWC and NFT currently comprise the majority of hydroponic systems utilized for leafy green production, additional systems are used, as illustrated in Figure 1. To our knowledge, little to no research has specifically been published on these lesser-known hydroponic systems. However, continuous drip and flood and drain systems are essentially modifications of DWC with the addition of a solid phase matrix and slight differences in how the water is circulated. Although not a commercial scale representation of either DWC-like systems, Kutter et al. [46] utilized quartz sand as a solid phase matrix in combination with Hoagland's medium for the germination and cultivation of barley (*Hordeum vulgare* var. Barke) in large, glass tubes. Here, microorganisms were introduced to the cultivation system by root-inoculation via the quartz sand matrix. While barley is not a leafy green, the study authors demonstrated the colonization and internalization of the plant shoot (stem and

leaves) with *S.* Typhimurium after four weeks. In contrast to the other studies highlighted in Table 2, Kutter et al. [46] inoculated the solid phase, although it is plausible to assume that microorganisms that had been inoculated in the nutrient solution would migrate to the sand matrix.

Moriarty et al. [57] also utilized a DWC-like system containing vermiculite in transplant trays. In this design, foam trays filled with a vermiculite mixture were directly seeded, and the trays were submerged in a tank of hydroponic nutrient water inoculated to a final concentration of 5 log CFU/mL. Holes at the base of the tray compartments allowed water to passively enter. Mean internalization for roots with no cut, two cuts, and three roots cuts $2.4 \pm 0.7$ CFU/g, $4.0 \pm 1.9$ CFU/g, and $3.3 \pm 1.3$ CFU/g, respectively. Carducci et al. [49] provided a similar system design to Moriarty et al. [57], and demonstrated the internalization of enteroviruses in lettuce leaves via nutrient solution contaminated with viruses. However, Carducci et al. [49] did not investigate the impact of damaged roots on the level of internalization. The impact of root damage is discussed further in Section 6.2.

An additional study investigated the internalization of *S.* Thompson and MNV into the edible parts of basil via the roots [52]. Here, the authors used a four-pot hydroponic drip system filled with clay pebbles. Basil seeds were germinated in peat pellets and then transplanted to the drip system. At six weeks old, basil plants in the peat pellets were removed from the pots and soaked in an inoculum of either MNV or *S.* Thompson for 1 h. Li and Uyttendaele [52] reported varying levels of MNV internalization on days 1 and 3 post-inoculation and positive *S.* Thompson on days 3 and 6 following sample enrichment. This study presents unique differences from the previously discussed research utilizing DWC-like systems. Most notable is the inoculation method directly to the plant roots via inoculum-soaked germination discs, as opposed to within the hydroponic nutrient water. While this may be analogous to nutrient water interactions with solid matrices, additional research specifically addressing the role of solid matrices in pathogen internalization by leafy greens is warranted.

The studies presented in Table 2 also encompass those that utilize an experimental setup lacking any representation of real-world hydroponic systems. Dong et al. [54] evaluated the rhizosphere and endophytic colonization of alfalfa (*Medicago sativa*) and barrelclover (*M. truncatula*) sprouts by enteric bacteria. Germinated seedlings with ~5 mm roots were transplanted into test tubes containing 10 mL of Jensen's nitrogen-free medium with 0.45% agar followed by inoculation of the medium (i.e., proximal to the seedling root area) 24 h later with prepared bacterial suspensions. Overall, endophytic colonization was observed for all of the enteric bacteria strains, with *Klebsiella pneumoniae* being the most efficient, and *E. coli* K-12 (generic strain) being the least efficient. The efficiency of all the *Salmonella* serovars and *E. coli* O157:H7 settled somewhere in the middle with respect to colonization abilities. For instance, a single CFU of *Salmonella* Cubana and Infantis inoculated to the root area resulted in interior colonization of alfalfa within five days post-inoculation, thus suggesting that no level of contamination is free of risk. Another primary observation from Dong et al. [54] was the correlation between endophytic and rhizosphere colonization. More specifically, the authors showed that as the colonization of the rhizosphere increased, there was a complimentary increase in the endophytic colonization of alfalfa by all of the bacterial strains ($r^2 = 0.729$–$0.951$) except for *E. coli* K-12 ($r^2 = 0.017$) [54].

Jablasone et al. [28] also utilized a hydroponic agar system to investigate the interactions of *E. coli* O157:H7, *S.* Typhimurium, and *L. monocytogenes* with plants at various stages in the production cycle. While the authors reported on two cultivation study designs, our focus will be on the cultivation studies lasting >10 days in which contaminated seeds were cultivated in 500-mL polypropylene flasks containing hydroponic solution solidified with 0.8% (w/v) agar. Here, the seeds—seven different plant types, including cress, lettuce, and spinach—were directly inoculated with pathogens (3.3 to 4.7 log CFU/g) and then germinated. Overall, pathogen levels increased significantly during the 10-day germination period. With respect to internalization, *S.* Typhimurium was detected in lettuce seedlings at nine days, but not thereafter, and *E. coli* O157:H7 was detected in lettuce and spinach seedlings also at nine days. Meanwhile, *L. monocytogenes* was not detected in the internal tissues of the

seedlings at any time point. Overall, the authors concluded that there seemed to be an age-specific effect on pathogen internalization, with younger plants being more susceptible. In addition, there were apparent plant-specific and pathogen-specific effects observed, with the latter also observed by Kutter et al. [46] with respect to the lack of internalization of *L. monocytogenes*, while other pathogens such as *E. coli* and *Salmonella* were internalized.

As alluded to in Section 5.1, the study by Klerks et al. [55] also utilized a hydroponic agar system to study the plant and microbial factors that impact the colonization efficiency of five *Salmonella* serovars with three commercially relevant lettuce cultivars (Cancan, Nelly, and Tamburo). Within the same study, the authors investigated the association of *Salmonella* with lettuce Tamburo grown in soil. For soil-based studies, only one serovar (Dublin) was detected in the plant tissue of lettuce Tamburo with a concentration of 2.2 log CFU/g. Meanwhile, *S.* Dublin, Typhimurium, Enteritidis, Newport, and Montevideo internalized in Tamburo at 4.6 CFU/g, 4.27 CFU/g, 3.93 CFU/g, ~3 CFU/g, and ~4 log CFU/g when cultivated hydroponically, respectively. Interestingly, while the prevalence of *Salmonella* in lettuce plant tissues was not impacted by the lettuce cultivar, there was a significant interaction between *Salmonella* serovar and cultivar with respect to the level of endophytic colonization (CFU/g) during hydroponic cultivation. Klerks et al. [55] further demonstrated the active movement of *S.* Typhimurium to the plant roots of lettuce Tamburo when placed in microcapillary tubes with root exudates, as well as the upregulation of pathogenicity genes. More specifically, the authors identified an organic compound in the root exudates that is used as a carbon source by *Salmonella* and observed the initiation of processes that allow for host cell attachment [55].

## 6. Targeted Preventive Controls in Hydroponic Systems for Leafy Vegetables

### 6.1. Production Water Quality and Whole System Decontamination

6.1.1. Current Agricultural Water Quality Guidelines for Fresh Produce

Since water is central to hydroponic plant culture, maintaining microbial water quality should be a primary control point for food safety. Guidelines for pre-harvest agricultural water have been put forth by the Food and Drug Administration (FDA) through the Food Safety Modernization Act (FSMA) and the Produce Safety Rule (PSR) (21 CFR § 112.42). Specifically, water used during growing activities must meet a geometric mean of ≤126 CFU/100 mL generic E. coli and a statistical threshold value of ≤410 CFU/100 mL generic E. coli based on a rolling four-year sample dataset. However, as with most aspects of the PSR, requirements are based on field-grown raw agricultural commodities without consideration for hydroponic systems. This raises the question of whether pre-harvest agricultural water standards should remain the same or be more or less stringent for hydroponic production. For instance, Allende and Monaghan [64] suggest hydroponic systems as a risk reduction strategy for leafy green contamination, as the water does not come into contact with the edible parts of the crop. However, this review has shown evidence to the contrary. Clearly, based on the data presented in this review, this is not a simple question given the differences in pathogen internalization across hydroponic system types as well as plant cultivars and pathogen strain type.

6.1.2. Risk of System Contamination

While maintaining high nutrient solution quality and preventing root damage are major factors in preventing internalization in leafy greens, a clean hydroponic system can prevent microorganisms from disseminating throughout the plant and beyond. For instance, Wang et al. [51] introduced MNV into their experimental NFT system to determine the internalization and dissemination of the virus in microgreens, as described in Section 5.2. After harvesting the microgreens on day 12, the remaining microgreens, hydroponic growing pads, and nutrient solution were removed without further washing or disinfection of the system. To start the new growth cycle, a new set of hydroponic growing pads and microgreen seeds were utilized for germination. Fresh nutrient solution was used, and no MNV

was inoculated. Even still, MNV was detected in the nutrient solution for up to 12 days (2.26 to 1.00 log PFU/mL) during this second growing cycle and was also observed in both the edible tissues and roots of the microgreens.

In a brief review of the microbial composition of hydroponic systems in the Netherlands, Waechter-Kristensen et al. [65] reported *Pseudomonas* spp. as the dominant species, with most of the total aerobic bacteria attached to gutter, growth substrate, and plant roots. In a more sophisticated analysis, Lopez-Galvez et al. [66] assessed two hydroponic greenhouse water sources for generic *E coli* as well as the pathogens *Listeria* spp., *Salmonella enterica*, and STEC. The authors found that generic *E. coli* counts were higher in reclaimed water than in surface water. Interestingly, *Listeria* spp. counts increased after adding the hydroponic nutrients in both surface and reclaimed water, although neither source showed significant differences in generic *E. coli* counts. STEC was not identified in any sample, but 7.7% of the water samples tested positive for *Salmonella* spp., and 62.5% of these were from the reclaimed water source. Regardless, the microbial contamination of nutrient solution did not translate into contaminated produce in this instance, as none of the tomato samples tested were positive for target microorganisms.

Another consideration is the impact of hydroponic feed water recirculation on pathogen survival. Routine system-wide water changes in hydroponic systems are likely costly and labor-intensive. As a result, hydroponic practitioners typically monitor nutrient levels in real time or by routine sampling and add nutrients and water as needed due to uptake and evaporation, respectively. Therefore, the need arises for routine microbiological testing of feed water and preparing nutrient solutions with treated water to prevent the rapid spread of pathogens through systems. Furthermore, there are no formal guidelines for how often to drain nutrient solution to waste and replace, rather than replenish as needed, other than the obvious scenarios following plant disease outbreaks [39]. Research is needed to demonstrate if such labor-intensive practices would have a beneficial effect on food safety in hydroponic systems.

### 6.1.3. Water Treatment Strategies

Methods for the continuous control of microbial water quality in recirculating hydroponic systems almost exclusively focus on the removal of plant pathogens and include membrane filtration [67], slow sand filtration, [68–71], and ultraviolet (UV) light treatment [72–74]. Methods for pre-treating water that are used to prepare nutrient solutions include ozonation [75], chlorination, iodine, or hydrogen peroxide. Biological control agents are also used [76] and are discussed further in Section 6.3. Each of these methods possesses advantages and disadvantages with respect to their practical use [72,77,78], as outlined in Table 3.

While ozone is a proven water treatment strategy [79], some investigators have suggested [71,77] that the ozonation of hydroponic nutrient water may lead to the precipitation of mineral nutrients such as manganese and iron due to the strong oxidizing properties of ozone. However, Ohashi-Kaneko et al. [75] found that the initial growth of tomato plants supplied with a nutrient solution prepared with ozonated water at a dissolved ozone concentration of 1.5 mg/L was greater than in non-ozonated water, indicating that ozonation is not only safe for young plants, but possibly beneficial. This is the most vulnerable stage for hydroponic vegetables and leafy greens, indicating that ozonation is a promising strategy particularly to prevent internalization at germination and early stages of growth.

Recently, Moriarty et al. [57] demonstrated that UV light successfully reduced natural levels of total coliforms by 3 log CFU/mL in nutrient water in a pilot-scale DWC aquaponics system. Moreover, lettuce samples were surface-sterilized using UV light in a biosafety cabinet as well as a bleach/detergent mixture prior to testing for internalized coliform bacteria, of which none were detected. Moriarty et al. [57] stated that this neither confirms nor refutes the effectiveness of UV light in preventing coliform internalization by lettuce in DWC aquaponics in an open environment. Nevertheless, the reduction of total coliforms in nutrient water is a desirable outcome and may be included in prevention guidelines if these effects can be replicated.

**Table 3.** Water treatment strategies and associated advantages and disadvantages.

| Method | Advantages | Disadvantages |
|---|---|---|
| Membrane filtration | Precise filtration, can choose pore size to suit needs | Reduced flow rate, easy clogging |
| Slow sand filtration | Most common, inexpensive, a variety of substrate choices. | May not effectively remove pathogens on its own |
| UV light treatment | Can be combined with slow sand filtration for high efficiency | Water needs high clarity, so must be combined with sediment filter to ensure maximum light penetration |
| Chlorination | Inexpensive, standard recommendation | Storage issues, toxic to humans |
| Iodine | Less toxic than chlorine | Need high doses to be effective, costly |
| Hydrogen peroxide | Less toxic than chlorine, weak oxidizer | Need high doses to be effective, costly |
| Ozonation | Non-toxic to humans, no residues left behind | Strong oxidizer may cause hydroponic mineral nutrients to precipitate, reducing bioavailability |
| Biological control agents | Takes advantage of natural features of the system to suppress pathogens without addition of harsh chemicals | Inconsistent, difficult to maintain microbial numbers to sufficiently suppress pathogens, manipulation of microbiome for this purpose still a poorly understood research area. |

## 6.2. Minimizing Root Damage

Damage to root tissue has been suspected to increase pathogen internalization in soil cultivation of leafy greens, but multiple reviews of current evidence suggest that only damage at root tips and lateral root junctions increases internalization under experimental conditions [7,30,48]. Similarly, root damage in most hydroponic studies are experimenter-induced. These bench scale investigations demonstrate that to some extent, root damage is linked to increased internalization in hydroponics as well. However, it is not known if incidental damage is more likely to occur in hydroponic systems or soil.

As discussed in Section 5.3, Moriarty et al. [57] demonstrated that intentionally severing root tips did increase E. coli O157:H7 internalization in deep water cultivated lettuce compared to uncut controls. While two cuts did increase internalization in a hydroponic system over uncut roots, adding a third cut did not show a statistically significant increase in internalization. Similarly, within a DWC cultivation system inoculated with 7 log CFU/mL of E. coli TG1, bacterial density was greater after 48 h in the shoots of corn seedlings with the entire root system removed (430 CFU/g) and with the root tips severed (500 CFU/g) compared to undamaged plants (18 CFU/g) [29]. These findings are similar to those in soil-based studies.

Guo et al. [80] utilized a DWC system and reported internalization of Salmonella serovars (Montevideo, Poona, Michigan, Hartford, Enteritidis) in the leaves, stems, hypocotyls, and cotyledons of tomato plants with both damaged and undamaged roots. The initial inoculum level was 4.46 to 4.65 log CFU/mL, and at nine days post-inoculation, Salmonella serovars remained between 3.5–4.5 log CFU/mL. Interestingly, internalization was greater in undamaged root systems when compared to damaged roots.

## 6.3. Biological Control

Since many hydroponic system designs involve the recirculation of nutrient water, the risk of pathogen spread via water in these systems has attracted considerable attention. The rapid advancement of next-generation sequencing technologies in recent years has spawned a research effort to characterize the microbiome of "-ponics" systems and to use this information to develop "probiotic" disease prevention strategies. Most of this work has been focused on the prevention of plant pathogens because of their direct impact on crop yield [81]. It is reasonable to assume that pathogens, where the plant is the natural host, will respond differently to biological control treatments

compared to pathogens that primarily infect humans. Nevertheless, a few studies have demonstrated a proof of concept that the introduction of putatively beneficial microorganisms has a noticeable effect on the plant microbiome, of which pathogens may or may not be a part [81–84].

Thus far, it has been demonstrated that the addition of beneficial bacteria or fungi to hydroponic systems may improve plant growth in some cases, either indirectly by the suppression of diseases such as root rot [85] or by improving nutrient bioavailability and uptake by altering the rhizosphere [86]. In other cases, the biological control gave mixed results. For example, Giurgiu et al. [87] found that *Trichoderma* spp. acted as a growth promoter, but not a disease suppressor. Although not purposely a study on bioinoculation, Klerks et al. [55] hypothesized the difference in the internalization of *Salmonella* in lettuce grown in soil versus axenically in a hydroponic agar-based system. More specifically, the authors suggest that the lack of endophytic colonization in soil-grown lettuce was due to the presence of native rhizosphere bacteria, and conversely, the absence of bacteria in the axenic system enabled *Salmonella* easier access to the roots.

Despite a growing body of research on plant protection, there are currently no studies on the use of beneficial bacteria or fungi to suppress the growth of human pathogens in and on crops in hydroponic systems. The biological control of fish and plant pathogens has been attempted in aquaponics [88]. Of the 924 bacterial isolates from the aquaponics system itself, 42 isolates were able to suppress the plant disease *Pythium ultimum* and fish oomycete pathogen *Saprolegnia parasitica* in vitro. Such interventions have not yet been tested in either bench-scale or larger hydroponic systems.

### 6.4. Plant Cultivar Selection

A few studies presented in this review have demonstrated the difference in pathogen internalization and colonization across plant cultivars, which raises the question as to whether cultivar selection could be a preventive control for the leafy vegetable hydroponics industry. As previously discussed in Section 5.3, Klerks et al. [55] demonstrated an interaction between the level (i.e., CFU/g leaf) of endophytic colonization of *Salmonella* and lettuce cultivar during hydroponic cultivation. Moreover, Klerks et al. demonstrated a specific interaction of *Salmonella* with root exudates from cultivar Tamburo, suggesting chemotaxis of *Salmonella* to the roots, and thus further aiding internalization. Another hydroponic agar system study [28] reported differences in the microbial colonization of the endophyte, although these differences were across plant genera and not cultivars within a specific species; even still, the authors demonstrated a plant-specific effect on the internalization of bacteria.

Meanwhile, although not based on a hydroponic cultivation system, Erickson et al. [89] investigated the ability of *Salmonella* to internalize in seven cultivars of leafy greens and one cultivar of Romaine lettuce. The authors spray-inoculated the foliage of three-week old transplants with green fluorescent protein (GFP)-labeled *Salmonella* (Enteritidis and Newport) and evaluated internalization at 1 and 24 h post-inoculation (p.i.). Simultaneously, non-inoculated plants were analyzed for total phenols and antioxidant capacity. Erickson et al. reported cultivar as a significant variable for the internalization of *Salmonella* via contaminated foliage. More specifically, leafy green cultivar Muir was the most likely to show endophytic colonization 1 h and 24 h p.i. Interestingly, there was an inverse relationship between the concentration of antimicrobials (i.e., phenols and antioxidants) and internalization prevalence, suggesting the importance of plant defenses against human pathogenic bacteria. However, overall, the path toward risk-based preventive controls based on cultivar selection in hydroponic production needs further investigation.

### 7. Potential Actual Health Risk from Consumption of Leafy Vegetables with Internalized Pathogens

While this review has focused on the risk of pathogen internalization in leafy vegetables grown hydroponically, how does this translate to actual human health risk? To begin, determining the specific health risk from internalized pathogens in leafy vegetables as opposed to contamination in general

is difficult. Clearly, there is a risk of illness regardless of where the pathogen is located on the edible portion of the leafy vegetable; however, the primary concern with respect to internalized pathogens is the inability to inactivate through post-harvest disinfection practices, as stated previously in this review (Section 3). As purported by Saper [90], one of the major limiting factors in decontamination efficacy includes the internalization of microbial contaminants within plant tissues, which basically precludes effective disinfection by washing or sanitizing agents.

Another aspect to consider is the infectious dose linked to the primary pathogens of concern for leafy vegetable contamination. *L. monocytogenes*, STECs, *Salmonella*, and human enteric viruses have all been documented to cause illness with as few as 10 to 100 infectious units (i.e., bacterial cells or virus particles) [91,92]. On the other hand, there exists extreme variability across strains of specific pathogens with respect to the estimated dose and resulting response (i.e., gastroenteritis). Based on the variable infectious dose as well as the average serving size of leafy vegetables (i.e., 1 to 2 cups, or approximately 75 to 150 g) [93] and the data reported in Table 2, the risk of becoming ill from the ingestion of leafy vegetables with internalized pathogens is highly probable in the event of gross levels of contamination. Unfortunately, the microbial load that is internalized under natural growing conditions has not been well-characterized. For example, in the event of a foodborne disease outbreak linked to leafy vegetables, not only is it rare to have product left to test, but if the pathogen of concern is detected, then whether the contamination was external or internal is not usually determined. Moreover, host factors including age, immune status, and gastrointestinal characteristics (e.g., stomach acid levels, commensal bacteria, immune cells) also play a critical role in the required infectious dose.

## 8. Conclusions

This review aimed to highlight the risks associated with human pathogen internalization in leafy vegetables cultivated in lab-scale hydroponic systems. The studies presented within this review (Table 2) overwhelming suggest that human pathogens—both viruses and bacteria—are readily internalized within plant tissues via the uptake of contaminated nutrient solution through the root system. The data also demonstrate the immense amount of variability in the hydroponic system setup, bacteria and virus type selection, method of inoculation, and plant cultivar selection, as well as techniques for the recovery and detection of microorganisms within plant tissues.

With respect to the recovery and detection of microorganisms, there are few differences that can be mentioned. For instance, Warriner et al. [50] utilized non-pathogenic, bioluminescent *E. coli* P36 for detection by fluorescence imaging as well as the β-glucuronidase (GUS) assay, where the gene for the enzyme β-glucuronidase was used as a reporter to measure cell viability and distribution. Sharma et al. [32] tested three strains of genetically engineered GFP-expressing *E. coli* O157:H7 detected by immunofluorescence. Additionally, not all investigators performed a leaf surface sterilization prior to microbial detection to rule out epiphytic bacteria [46,47,52]. However, the natural contamination of bacteria at significant levels is unlikely due to the high inoculation levels of the specific strains used in the study combined with the aseptic environment of lab-scale systems. Furthermore, surface sterilization protocols vary widely, and may be differentially effective.

As hydroponic systems, particularly DWC, continue to increase in popularity, the impact of plant cultivar, system type, and microbial type/strain on microorganism internalization needs further characterization. In order to further the knowledge and understanding within this specialized research area, several recommendations for the standardization of research related to hydroponic cultivation of leafy vegetables for the investigation of interactions with human pathogens have been provided:

- Development of standard guidelines for lab-scale hydroponic cultivation of leafy vegetables to enable study comparison. This includes seed germination protocols, best practices for water management, and design specifications for each type of hydroponic system.
- Determine appropriate pathogen inoculation concentrations and methods for the research question being addressed. Should there be a range of concentrations considered? How does

the inoculation of the seed at germination versus inoculation of the nutrient solution change the interpretation of the results?

- Does the presence of a solid substrate impact colonization efficiency? Is there a differential effect between contamination of the substrate and the contamination of nutrient water flowing through it?
- Standardization of microbial extraction methods from plants to ensure the recovery of truly endophytic microorganisms.
- Selection of microorganisms should be standardized. For instance, surrogate microorganisms should be validated as representative of their human pathogen counterparts. Strains of human pathogens should also be carefully considered and validated for use in hydroponic cultivation systems.
- Given the variation in the susceptibility of plants to pathogen colonization, the selection of plant cultivars should be standardized to represent commercially relevant cultivars, and the validation of cultivars used in hydroponic research is needed.

**Author Contributions:** All of the authors contributed equally to the conception, writing, and final review of the manuscript.

**Acknowledgments:** This research was supported in part by the National Institute of Food and Agriculture (NIFA), U.S. Department of Agriculture (USDA), Hatch Act.

**Conflicts of Interest:** The authors declare no conflict of interest.

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
