# Peer review of "Risk of Human Pathogen Internalization in Leafy Vegetables During Lab-Scale Hydroponic Cultivation"

_horticulturae, doi:10.3390/horticulturae5010025_

Round 1

Reviewer 1 Report

This manuscript provides a review of the various types of hydroponic systems with regard to the internalization of human enteric pathogens in hydroponically grown fresh produce.  It also addresses points where preventive controls in those hydroponic systems could be applied to reduce contamination of fresh produce (including leafy greens) with pathogens.  The manuscript is well written with appropriate organization of the information.  The types of hydroponic systems and potential risks of produce contamination associated with those systems are discussed with ample reference to pertinent published research. 

While it is acceptable that surface contamination of fresh produce with pathogens poses a strong foodborne disease risk to consumers, the risk posed by internalized pathogens need to be addressed from a “real-world” perspective.  The authors have addressed the issue of pathogen internalization into produce grown in hydroponic systems.  However, in the pathogen internalization section, the manuscript “begs” for a brief discussion on the health risk that pathogens internalized pathogens pose to consumers.  In this respect, the following are some relevant questions:  Under natural conditions, have internalized pathogens been reported at levels that represent an infectious dose?  Considering the average amount of leafy salad vegetables an average person is likely to consume, could this be a potential health risk if an internalized pathogen is one with a low infectious dose?  Could we speculate that pathogens internalized in fresh produce will be protected from the hostile environment of the human digestive tract thus increasing foodborne disease risk in consumers?

In this reviewer’s opinion addressing these previously stated issues would be beneficial to readers and strengthens the importance of the microbial food safety aspect of this manuscript.  Please see the following minor comments:

Line 255: "..inherit plant defenses".  The word "inherit" is a verb that is being used as an adjective in this writing.  Please replace it with an appropriate word.  This reviewer's suggestion is to use the word "inherent" if the authors agree with this suggestion.

Line 388: Change "demonstrate" to "demonstrated" to maintain the consistent tense through out the manuscript.

Author Response

Reviewer 1

Comment: While it is acceptable that surface contamination of fresh produce with pathogens poses a strong foodborne disease risk to consumers, the risk posed by internalized pathogens need to be addressed from a “real-world” perspective. The authors have addressed the issue of pathogen internalization into produce grown in hydroponic systems. However, in the pathogen internalization section, the manuscript “begs” for a brief discussion on the health risk that pathogens internalized pathogens pose to consumers. In this respect, the following are some relevant questions: Under natural conditions, have internalized pathogens been reported at levels that represent an infectious dose? Considering the average amount of leafy salad vegetables an average person is likely to consume, could this be a potential health risk if an internalized pathogen is one with a low infectious dose? Could we speculate that pathogens internalized in fresh produce will be protected from the hostile environment of the human digestive tract thus increasing foodborne disease risk in consumers?

Response: Thank you for the suggestion.  We agree that this section of potential risk is warranted, and we have added a section (new Section 7) in the revision to address this comment (Line 578-602)

Line 255: "..inherit plant defenses". The word "inherit" is a verb that is being used as an adjective in this writing. Please replace it with an appropriate word. This reviewer's suggestion is to use the word "inherent" if the authors agree with this suggestion.

Response: Yes, this was a mistake. We meant to say “inherent”, and this has been corrected in the revision.

Line 388: Change "demonstrate" to "demonstrated" to maintain the consistent tense throughout the manuscript

Response: We have made this correction.  

Reviewer 2 Report

In the review article, the authors talked about different hydroponic system designs in the light of pathogen internalization in leafy greens. Overall, the article is well written, the topic is interesting. The authors need the revise the manuscript text for grammatical errors, elaborate abbreviations when used for the first time, provide appropriate citations whenever necessary.

The following recommendations may further improve the article:

Was it a systematic review? The authors need to describe whether their review article covers all the recent related publications by year and the key words that were used to search for the articles. If not systematic review, the authors need to discuss how the relevant literature was sorted, how much information was omitted in the article? Introduction needs a statement on how the study may be novel.

Line 27: “reluctantly identified” phrase was not used properly. Please rephrase the statement.

Line 35: “CFA is a growing industry” could the authors provide an approximate size of the industry (or some historical growth data) in the US?

Lines 59-62: Is there any reference for hydroponic leafy green production system categories? Are these six categories comprehensive? The authors need to validate the completeness different hydroponic systems.

Lines 66-67: Needs citation. And what does the author mean by, “the next five years”. Need to specify the dates.

Line 73: The authors mentions about “numerous studies” yet cites only one reference.

Line 95: “103- to 105-fold” the information is correct?

Table 1 and Figure 1: The authors did not include “aeroponics” in the comparison. In addition, it would be beneficial for the readers if the authors included some references that described each of the designs in depth.

Line 184-186: “C. freundii” genus names should be elaborated when used in the article for the first time. The authors need to further discuss on the effects of surface sterilization on the pathogen survival.

Table 2: The authors need to summarize this table or add a new summarized table that compare all the hydroponic systems in the light of pathogen internalization. The authors discussed this in the text, but a tabular representation of the information will be more reader friendly.

Line 229: Which serovar was more prone to internalization?

Line 561: “high inoculation levels” is it correct?

Author Response

Reviewer 2

Comment: The authors need the revise the manuscript text for grammatical errors, elaborate abbreviations when used for the first time, provide appropriate citations whenever necessary.

Response:  The revision has been proof-read for grammar and acronyms.

Comment: Was it a systematic review? The authors need to describe whether their review article covers all the recent related publications by year and the key words that were used to search for the articles. If not systematic review, the authors need to discuss how the relevant literature was sorted, how much information was omitted in the article? Introduction needs a statement on how the study may be novel.

Response:  We have added subsection 1.1 to explain the review methodology as well as the novelty (Line 52-69).

Line 27: “reluctantly identified” phrase was not used properly. Please rephrase the statement.

Response: The phrase has been changed to “was implicated in.”

Line 35: “CEA is a growing industry” could the authors provide an approximate size of the industry (or some historical growth data) in the US?

Response: We have added some industry growth statistics that are based on a survey by a start-up called Agrylist that sells indoor farming management software. It is not an academic survey, but one of the few available that has kept track of industry growth between 2016 and 2017. Other than this, the industry is not well-characterized for growth in total acreage over time.

Lines 59-62: Is there any reference for hydroponic leafy green production system categories? Are these six categories comprehensive? The authors need to validate the completeness different hydroponic systems.

Response: We have added a reference which describes all of the systems we focused on in our review. To our knowledge, it is one of the first review articles to separate hydroponics by system type and has been cited 102 times: Jensen, M.H. Hydroponics Worldwide. Acta Hortic. 1999, 481, 719–729.

Additionally, a more recent reference describes the substrates used in hydroponics: Asaduzzaman, M.; Saifullah, M.; Mollick, A.K.M.S.R.; Hossain, M.M.; Halim, G.M.A.; Asao, T. Influence of Soilless Culture Substrate on Improvement of Yield and Produce Quality of Horticultural Crops. In Soilless Culture - Use of Substrates for the Production of Quality Horticultural Crops; InTech, 2015.

Lines 66-67: Needs citation. And what does the author mean by, “the next five years”. Need to specify the dates.

Response: We have added the Agrylist 2016 survey reference to be clear that this sentence was informed by that reference. Additionally, we added a clarifying phrase that “over the next five years” was from the survey question that was asked – “Are you planning to expand your farm over the next five years?” Specific dates were not provided because it was not asked that way, and somewhat subjective in relation to each individual grower’s plans.

Line 73: The authors mention about “numerous studies” yet cites only one reference.

Response: We have added a clarifying phrase to indicate that the numerous studies have already been reviewed extensively by De Keuckelaere and co-authors (2015). (Line 93-96)

Line 95: “103- to 105-fold” the information is correct?

Response: We have revised the sentence as follows “Hara-Kudo and co-authors demonstrated that contamination of either the seed or hydroponic water with E. coli O157:H7 can result in marked colonization of the edible parts of the sprout. In addition, frequency of internalization increased with increasing concentrations of E. coli O157:H7 in the hydroponic water” (Line 116-119)

Table 1 and Figure 1: The authors did not include “aeroponics” in the comparison. In addition, it would be beneficial for the readers if the authors included some references that described each of the designs in depth.

Response: A statement has been added on Lines 17, 82, and 156-159 that aeroponics will not be discussed in this review.

Line 184-186: “C. freundii” genus names should be elaborated when used in the article for the first time. The authors need to further discuss on the effects of surface sterilization on the pathogen survival.

Response: We have checked the document and have spelled out the genus names of microorganisms upon first mention.  With respect to surface sterilization, assuming that you are referring to the importance of this step in determining internalization, we have added additional information regarding the use of surface decontamination in internalization studies (Line 199 to 205): “Briefly, Table 2 is designed to highlight key aspects impacting microbial internalization results of the lab-scale hydroponic studies including type of microorganisms, plant type and cultivar, inoculation procedure, and the application of surface sterilization prior to microbial analysis. With respect to surface sterilization, 12 out of the 17 studies cited in Table 2 specifically described application of a decontamination procedure prior to microbial recovery and detection. Most investigators validated decontamination procedures and showed complete inactivation of external microorganisms while maintaining viability of internalized microorganisms.”

Table 2: The authors need to summarize this table or add a new summarized table that compare all the hydroponic systems in the light of pathogen internalization. The authors discussed this in the text, but a tabular representation of the information will be more reader friendly.

Response: While we agree that it would be beneficial to rank the hydroponic system types by pathogen internalization risk, it is simply not possible given the multiple limitations that are presented in Section 5 as well as in light of the recommendations for standardization of internalization studies presented in the conclusions. Table 2 is designed not only to summarize the key studies that have been published, but also to highlight the lack of standardization and thus difficulty in making any overarching conclusions. In summary, we believe that it would be irresponsible to compare the pathogen internalization risk of the systems with the limited available data. However, we have added a few more descriptive details related to the table as a response to a previous comment (Line 199-205)

Line 229: Which serovar was more prone to internalization?

Response: As indicated in the text, this study was discussed in more depth in Section 5.3, Line 407 to 422. Here, it is stated that all serovars were internalized with S. Dublin having the highest concentration in lettuce cultivar Tamburo.  More importantly, the authors demonstrated a significant interaction between Salmonella serovar and cultivar with respect to endophytic colonization. So, the ability to internalize is not solely driven by the serovar.

Line 561: “high inoculation levels” is it correct?

Response: We agree that this was not clear.  We meant to distinguish between “natural” contamination of bacteria as opposed to inoculated. We have clarified the statement (Lines 617-619): “However, natural contamination of bacteria at significant levels is unlikely due to high inoculation levels of the specific strains used in the study combined with the aseptic environment of lab-scale systems.”